# AutoBasisEncoder: Pre-trained Neural Field Basis via Autoencoding for Operator Learning

**Thomas X Wang[1], Nicolas Baskiotis[1] & Patrick Gallinari[1, 2]**
[1] Sorbonne Université, CNRS, ISIR, 75005 Paris, France
[2] Criteo AI Lab, Paris, France
`thomas.wang@sorbonne-universite.fr`

## Abstract

We introduce AutoBasisEncoder, a novel framework designed for operator learning – the task of learning to map from one function to another. This approach autonomously discovers a basis of functions optimized for the target function space and utilizes this pre-trained basis for efficient operator learning. By introducing an intermediary auto-encoding task to the popular DeepONet framework, AutoBasisEncoder disentangles the learning of the basis functions and of the coefficients, simplifying the operator learning process. Initially, the framework learns basis functions through auto-encoding, followed by leveraging this basis to predict the coefficients of the target function. Preliminary experiments indicate that AutoBasisEncoder's basis functions exhibit superior suitability for operator learning and function reconstruction compared to DeepONet. These findings underscore the potential of AutoBasisEncoder to enhance the landscape of operator learning frameworks.

## 1 Introduction

Traditional approaches for solving PDEs often rely on predefined basis functions, limiting their adaptability to complex spatial patterns. Central to the success of these methods is the choice of basis, so that it can effectively represent the solution space. While traditional basis functions like Fourier series and polynomials have been widely employed, they usually have limited flexibility, due to being problem-specific (Meuris et al., 2023). Additionally, handcrafting an appropriate basis for a given problem often requires extensive domain knowledge.

Recently, Lu et al. (2021) introduced DeepONet, a framework that learns a basis of functions directly from data. They represent the target function as a linear combination of basis functions, which are learned jointly with a network that predicts the corresponding coefficients. To address the complexity of this complex high-dimensional optimization problem (Wang et al., 2022b), we explore decomposing the DeepONet scheme into two distinct, simpler sub-problems: optimizing the basis network and coefficient network separately.

This idea has been explored by POD-DeepONet (Lu et al., 2022), who apply a Proper Orthogonal Decomposition (POD) on discretized target functions, learning a basis that is then used to optimize the *branch* network. Notably, the outcomes of this approach are discretizations of functions on a pre-defined grid, thus requiring the incorporation of additional interpolation methods to obtain predictions outside of this grid (Witman et al., 2022; Demo et al., 2023; Meuris et al., 2023).

In this context, we propose AutoBasisEncoder, a two-step framework that breaks down DeepONet's optimization into two parts: the learning of basis functions and the learning of their coefficients. By introducing an intermediary auto-encoding objective, we first focus on learning a basis of functions tailored to our data; and leverage this pre-trained basis to learn the operator itself, by learning the coefficient network. The auto-encoding task helps to learn a basis of functions with better representation power.

Preliminary experiments show that this disentanglement of the DeepONet's optimization leads to the learning of a basis that captures the target function space better, and that this basis can improve the operator learning.

To summarize, AutoBasisEncoder simplifies the optimization of DeepONet through an additional auto-encoding task. The resulting pre-trained basis of functions is more expressive and easier to optimize for than the basis of DeepONet.

## 2 METHODOLOGY

### 2.1 PROBLEM SETTING

Our goal is to learn a mapping between two functional spaces.

Let $\mathcal{U} \subset L^2(\Omega_{\mathcal{U}}, \mathbb{R}^p)$ and $\mathcal{V} \subset L^2(\Omega_{\mathcal{V}}, \mathbb{R}^q)$ be two infinite-dimensional function spaces, with $\Omega_{\mathcal{U}}$, $\Omega_{\mathcal{V}}$ compact subsets of $\mathbb{R}^m$ and $\mathbb{R}^n$. Our goal is to learn an operator $\mathcal{F}$ that maps functions from space $\mathcal{U}$ to space $\mathcal{V}$:

$$\mathcal{F} : \mathcal{U} \to \mathcal{V}, \quad u \mapsto v$$

In practice, we are given $N$ pairs of functions $(u_i|_{\mathcal{X}_i}, v_i|_{\mathcal{Y}_i})_{i \in [\![1,N]\!]}$ discretized on grids $\mathcal{X}_i \subset \Omega_{\mathcal{U}}$ and $\mathcal{Y}_i \subset \Omega_{\mathcal{V}}$.

### 2.2 DEEPONET

AutoBasisEncoder's architecture is based on DeepONet (Lu et al., 2021). In the DeepONet framework, the target function $v$ is modeled as a linear combination $\boldsymbol{\lambda} = (\lambda_1, \ldots, \lambda_d)$ of a set of basis functions: $\boldsymbol{\Phi} = \{\Phi_i\}_i$, so that $\hat{v}_{\boldsymbol{\lambda}, \boldsymbol{\Phi}}(x) = \sum_{i=1}^d \lambda_i \Phi_i(x)$. The network responsible for predicting the coefficients $(\lambda_i)_i$ is referred to as the *branch net*, while the network handling the function bases is known as the *trunk net*.

As illustrated in Figure 1, DeepONet approximates the operator $\mathcal{F}$ with: $\hat{\mathcal{F}} = L_{\boldsymbol{\Phi}} \circ U$, where $L_{\boldsymbol{\Phi}}$ is a linear combination with a basis of function $\mathcal{B} = (\Phi_i)_i$, and $U$ is the coefficient predictor:

$$U : \mathcal{U} \to \mathbb{R}^d, \quad u \mapsto (\lambda_1, \ldots, \lambda_d),$$

$$L_{\boldsymbol{\Phi}} : \mathbb{R}^d \to \mathcal{V}, \quad (\lambda_1, \ldots, \lambda_d) \mapsto \sum_{i=1}^d \lambda_i \Phi_i,$$

**Challenges in DeepONet** While DeepONet aims to learn a basis for representing the output space $\mathcal{V}$, the primary focus of the operator learning task is not centered on learning an expressive basis, but rather on efficiently processing $u$ to accurately map it to $v$. Consequently, DeepONet is compelled to strike a delicate balance between basis learning and coefficient learning, which makes its optimization highly complex, potentially resulting in a compromise of basis expressivity. As introduced better, previous work such as POD-DeepONet (Lu et al., 2022) have shown that explicitly stating these two sub-tasks (basis learning and coefficient learning) as separate objectives can enhance the operator learning process.

### 2.3 AUTOBASISENCODER

Building upon this understanding, we introduce AutoBasisEncoder — a novel framework specifically designed to address these two sub-tasks separately. While AutoBasisEncoder shares the same network architecture as DeepONet, AutoBasisEncoder distinguishes itself through the incorporation of an intermediary auto-encoding task (see Figures 1 and 2). The goal of this task is to promote a basis of functions that better represents the output space $\mathcal{V}$. Consequently, AutoBasisEncoder disentangles the optimization process, breaking down the complex task of operator learning into two simpler tasks: basis learning and coefficient learning.

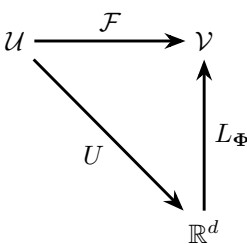

Figure 1: DeepONet decomposition. $L_\Phi$ is trained concurrently with $U$.

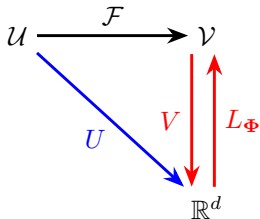

Figure 2: AutoBasisEncoder's Two-Step Decomposition: 1. Training $L_\Phi$ with $V$. 2. Freezing $L_\Phi$ and training $U$.

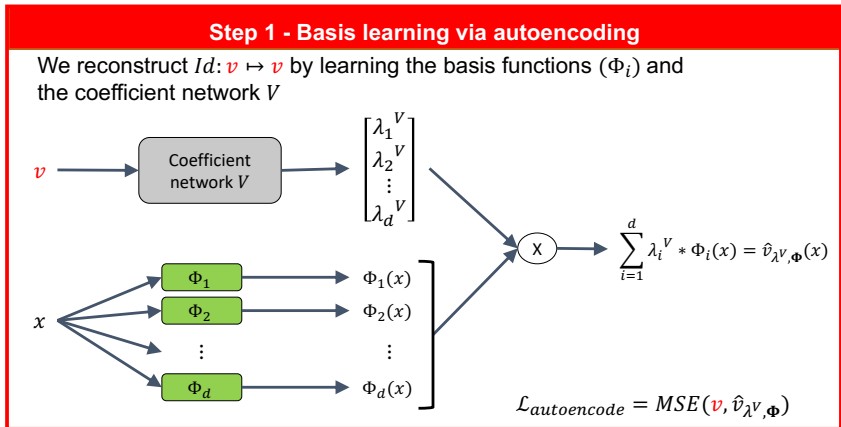

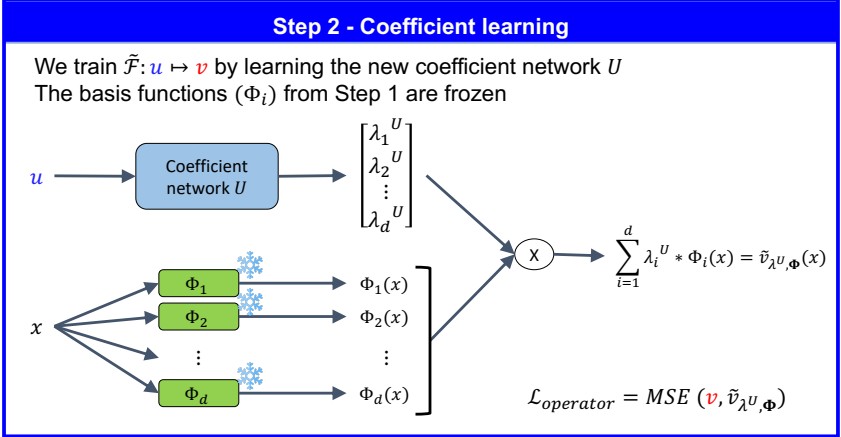

Figure 3: Detailed AutoBasisEncoder architecture. On top, the basis pre-training step: through the task of autoencoding $\mathcal{V}$, we learn the set of functions. $L_\Phi$ is trained concurrently with $U$. On the bottom, the coefficient learning step: the networks $\{\Phi_i\}_i$ are frozen, and only the coefficient network $U$ is trained.

**Pre-training a neural field basis of the output space**  We propose to discover a basis of functions via auto-encoding the target space $\mathcal{V}$. We constrain our latent space to represent the coefficients corresponding to the decomposition on this basis.

During this phase, both the basis of functions $\mathbf{\Phi}$ and the coefficient predictor $V : \mathcal{V} \to \mathbb{R}^d$ are learned concurrently, so that: $L_\Phi \circ V \approx Id_\mathcal{V}$. In other terms, for any function $v \in \mathcal{V}$, we auto-encode it into a vector $\boldsymbol{\lambda}^V = (\lambda_1^V, \dots, \lambda_d^V) \in \mathbb{R}^d$ so that: $\forall x \in \Omega_\mathcal{V}, \hat{v}_{\boldsymbol{\lambda}^V, \mathbf{\Phi}}(x) = \sum_{i=1}^d \lambda_i^V \Phi_i(x) \approx$

$v(x)$, where: $\boldsymbol{\lambda^V} = V(v)$. For supervision, we use a MSE loss between the groundtruth $v$ and its reconstruction: $\mathcal{L}_{autoencode} = MSE(v, \hat{v}_{\boldsymbol{\lambda^V},\boldsymbol{\Phi}})$. A detailed view of this step is presented in the top part of Figure 3.

**Coefficient Learning**   In its second phase, AutoBasisEncoder focuses on acquiring the coefficient network $U : \mathcal{U} \rightarrow \mathbb{R}^d$, where $u$ maps to $\boldsymbol{\lambda^U} = (\lambda_1^U, \ldots, \lambda_d^U) \in \mathbb{R}^d$. The objective is to ensure that the composition of the basis learning operator $L_{\boldsymbol{\Phi}}$ (which was pre-trained during the previous step) with the coefficient network $U$ approximates the true operator $\mathcal{F}$, expressed as $L_{\boldsymbol{\Phi}} \circ U \approx \mathcal{F}$, as shown in the bottom part of Figure 3. We denote $\hat{v}_{\boldsymbol{\lambda^U},\boldsymbol{\Phi}}(x) = \sum_{i=1}^{d} \lambda_i^U \Phi_i(x)$. This step's goal is to learn the operator, therefore we supervise this step via a MSE loss on the target function $v$: $\mathcal{L}_{operator} = MSE(v, \hat{v}_{\boldsymbol{\lambda^U},\boldsymbol{\Phi}})$. Unlike other operator methods (Li et al., 2021; Lu et al., 2022), which often require specific sampling constraints on the input functions, we do not assume specific architectures for processing the input function $u$, allowing all kinds of architectures as encoders.

## 3   EXPERIMENTS

We first compare the performance of AutoBasisEncoder on standard operator learning tasks, and then compare the bases learned by the different methods.

We evaluate the performance of AutoBasisEncoder and DeepONet over two standard datasets for operator learning: the Darcy flow equation and the Navier-Stokes equation dataset with viscosity $1e - 3$ from Li et al. (2021). Further details are presented in Appendix A.2.1.

### 3.1   PERFORMANCE

Our model is compared to vanilla DeepONet (Lu et al., 2021), and POD-DeepONet (Lu et al., 2022). For fairness, these methods all share the same architecture for the coefficient network, use 64 basis elements and are trained during 20k epochs in total, allowing convergence.

We introduce the two following tasks:

$\mathcal{T}_{autoencode}$ : Given an input function $v \in \mathcal{V}$, we aim to reconstruct $v$ itself.
$\mathcal{T}_{operator}$ : Given an input function $u \in \mathcal{U}$, we aim to predict the corresponding $v = \mathcal{F}(u) \in \mathcal{V}$.

As we can see in Table 1, AutoBasisEncoder improves vanilla DeepONet performance on both datasets, showcasing the importance of our pre-training step. We can see that our framework performs well on the Operator learning task, even though it was first trained on the auto-encoding task. It also performs better than POD-DeepONet on the *Darcy* dataset. POD-DeepONet outperforms AutoBasisEncoder on the *Navier-Stokes* dataset, but AutoBasisEncoder can provide continuous predictions all over the space $\mathcal{V}$, while POD-DeepONet can only produce predictions on points of its training data grid.

Table 1: Comparing learned bases via AutoBasisEncoder, DeepONet, and POD-DeepONet on different datasets. Metrics are reported in MSE ($\downarrow$). All numbers are expressed in $1 \times 10^{-4}$.

| Method | Darcy | | Navier-Stokes | |
|---|---|---|---|---|
| | $\mathcal{T}_{autoencode} \pm \sigma$ | $\mathcal{T}_{operator} \pm \sigma$ | $\mathcal{T}_{autoencode} \pm \sigma$ | $\mathcal{T}_{operator} \pm \sigma$ |
| DeepONet | N/A | $7.15 \pm 0.33$ | N/A | $4.92 \pm 2.49$ |
| POD-DeepONet | **1.30 $\pm$ 0.15** | $9.09 \pm 0.19$ | **1.20 $\pm$ 0.12** | **3.43 $\pm$ 0.30** |
| AutoBasisEncoder | $1.56 \pm 0.10$ | **6.22 $\pm$ 0.42** | $1.36 \pm 0.61$ | $4.02 \pm 1.95$ |

### 3.2   BASIS COMPARISON

In this section, we evaluate the efficacy of the acquired function bases by examining their ability to comprehensively span the target function space $\mathcal{V}$.

We evaluate the effectiveness of the bases learned by DeepONet, AutoBasisEncoder, and POD-DeepONet by assessing their ability to represent target function spaces $\mathcal{V}$. To do this, we first pre-

train the bases using each respective algorithm. Then, we use gradient descent to learn the optimal coefficients for linearly decomposing the target function space $\mathcal{V}$ in order to auto-encode it. These bases are trained on regular grids, with a resolution of 64 for the Navier-Stokes dataset and 29 for Darcy equation. We then use these bases to represent functions sampled at different resolutions: 32, 64, and 128. The results of this evaluation are summarized in Table 2. Notably, the basis produced by AutoBasisEncoder consistently outperforms that of DeepONet. Furthermore, AutoBasisEncoder also facilitates faster convergence of the gradient descent algorithm to the optimal coefficients, as demonstrated in Figure 5. While POD-DeepONet's basis enables better reconstruction of the function space $\mathcal{V}$, it is constrained by its discrete nature and is unable to accurately reconstruct functions with different discretizations from those used during training. Conversely, DeepONet and Auto-BasisEncoder exhibit greater flexibility, as they employ functions as components of their bases, allowing them to reconstruct functions sampled differently from the training sampling.

Table 2: A comparison of the expressivity of DeepONet and AutoBasisEncoder's basis functions

| Method | Darcy | Navier-Stokes | | |
|---|---|---|---|---|
| | | NS-64 | NS-128 | NS-32 |
| DeepONet | $1.32{\times}10^{-4}$ | $2.48{\times}10^{-5}$ | $3.79{\times}10^{-2}$ | $3.83{\times}10^{-2}$ |
| POD-DeepONet | $\mathbf{8.60{\times}10^{-5}}$ | $\mathbf{3.50{\times}10^{-6}}$ | N/A | N/A |
| AutoBasisEncoder | $1.07{\times}10^{-4}$ | $1.36{\times}10^{-5}$ | $\mathbf{3.08{\times}10^{-2}}$ | $\mathbf{3.16{\times}10^{-2}}$ |

Therefore, compared to DeepONet, AutoBasisEncoder's learned basis demonstrates better suitability for capturing the underlying structure of the output functions in space $V$, and while also improving the effectiveness in learning the operator $\mathcal{F}$.

However, a discernible gap exists between the MSE of operator learning and the MSE of the $\mathcal{V}$ space reconstruction task using AutoBasisEncoder's basis, indicating room for further improvement.

## 4 CONCLUSION

We propose AutoBasisEncoder, a novel framework that disentangles the operator learning task into two sequential tasks: the learning of an appropriate basis of neural networks that can represent the output function space more accurately than DeepONet, and the prediction of the target function via its corresponding basis coefficients. Through an intermediary auto-encoding task, this disentangling makes it possible to learn a basis of functions that is more suitable for multiple tasks, and in particular operator learning. This framework shows potential for operator learning towards better basis functions learning.

### ACKNOWLEDGMENTS

We acknowledge the financial support provided by DL4CLIM (ANR-19-CHIA-0018-01), DEEP-NUM (ANR-21-CE23-0017-02), PHLUSIM (ANR-23-CE23-0025-02), and PEPR Sharp (ANR-23-PEIA-0008", "ANR", "FRANCE 2030").

This work was performed using HPC resources from GENCI–IDRIS (Grant 2023-AD011013332R1).

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

# A APPENDIX

## A.1 RELATED WORK

**Auto-encoding in Operator learning** Works such as Oommen et al. (2022) and Li et al. (2023) utilize CNN encoder-decoder architectures for auto-encoding functions in a latent space. They then learn temporal evolution by fitting a DeepONet in this latent space. BelNet (Zhang et al., 2023) encode their input function via a POD-like decomposition, while Zhang et al. (2022) uses an auto-encoding framework based on DeepONet, specifically tailored for Stochastic Differential Equations.

One common challenge in these methods is the interpretability of the latent spaces. By adopting DeepONet as a general framework, we can view the predicted coefficients of the linear decomposition as a latent vector, which significantly enhances the interpretability of the latent space. This approach provides a more intuitive understanding of the latent space's content and facilitates insights into the learned representations.

Among existing methods, BasisONet (Hua & Lu, 2023) stands out as the closest to AutoBasisEncoder: BasisONet auto-encodes both the input and output spaces onto orthonormal bases of functions, and uses these auto-encoding constraints as additional regularizing loss terms. On the other hand, AutoBasisEncoder adopts a distinct approach by concentrating solely on auto-encoding the output function space, offering greater flexibility in processing the input function, enabling it to effectively handle non-continuous functions.

## A.2 EXPERIMENTS

### A.2.1 DATASETS

**Darcy equation** We consider the steady-state of the 2D Darcy Flow equation defined on the unit box, represented by the second-order, linear Partial Differential Equation (PDE):

$$-\nabla \cdot (a(x)\nabla u(x)) = f(x) \quad \text{for} \quad x \in (0,1)^2$$

subject to the Dirichlet boundary condition $u(x) = 0$ for $x \in \partial(0,1)^2$, where $a \in L^\infty((0,1)^2; \mathbb{R}^+)$ denotes the diffusion coefficient, and $f \in L^2((0,1)^2; \mathbb{R})$ represents the forcing function.

We aim to learn the operator $\mathcal{F} : L^\infty((0,1)^2; \mathbb{R}^+) \to H_0^1((0,1)^2; \mathbb{R}^+)$, which maps the diffusion coefficient $a$ to the solution $u$. It describes a flow fluid through a porous medium.

We use the dataset from Li et al. (2021), consisting of 1000 couples of functions, discretized on a 29x29 regular grid.

**Navier-Stokes equation** We investigate the behavior of a 2D Navier-Stokes equation on the unit torus. We consider the flow of a viscous, incompressible fluid. The equation involves the evolution of vorticity $w(x,t)$ over time, subject to certain boundary conditions and initial vorticity $w_0(x)$. Our objective is to learn an operator $\mathcal{F}$ that maps vorticity values $w(x,t)$ from a given time $t = 0$ to a later time $T$: $\mathcal{F} : w(x, t = 0) \mapsto w(x, t = T)$.

The 2D Navier-Stokes equation is describes as follows:

$$\begin{align}
\frac{\partial w(x,t)}{\partial t} + u(x,t) \cdot \nabla w(x,t) &= \nu\Delta w(x,t) + f(x), & x \in (0,1)^2, \quad t \in (0,T] \\
\nabla \cdot u(x,t) &= 0, & x \in (0,1)^2, \quad t \in [0,T] \\
w(x,0) &= w_0(x), & x \in (0,1)^2
\end{align} \tag{1}$$

where $u$ is the velocity field and $w = \nabla \times u$ is the vorticity. $w_0$ is the initial vorticity, $\nu \in \mathbb{R}^+$ is the viscosity coefficient of the fluid. A forcing function $f \in L^2_{\text{per}}((0,1)^2; \mathbb{R})$ is used.

We use the Navier-Stokes equation dataset with viscosity $\nu = 1e-3$ from Li et al. (2021). It consists of 1000 couples of vorticity functions at times 0 and $T$: $(w(x,t = 0), w(x,t = T))$, generated on a 256x256 regular grid, and then downsampled at a 64x64 resolution. We also use the 32 and 128 resolution data to assess the capabilities of our framework to model the distribution of vorticities $w(x,t = T)$ at time $T$.

### A.2.2 IMPLEMENTATION DETAILS

In practice, we instantiate our neural basis using neural networks with sinusoidal activation functions, capitalizing on their excellent high-frequency representation capabilities (Tancik et al., 2020; Sitzmann et al., 2020; Wang et al., 2022a). They indeed yield better performance than more traditional activation functions (ReLU, tanh...). Unlike the sparse Mixture of Experts employed by Wang et al. (2022a), we empirically found that a non-sparse mixture better suits our operator learning tasks. We use the $L^2$ norm for supervision during the basis learning step and during the coefficient learning step.

### A.2.3 VISUALIZATION OF RESULTS

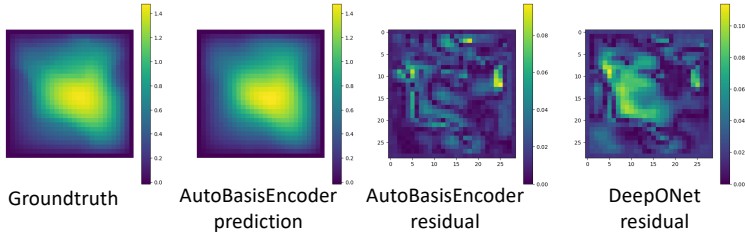

|            |                              |                            |                  |
|:----------:|:----------------------------:|:--------------------------:|:----------------:|
| Groundtruth | AutoBasisEncoder prediction | AutoBasisEncoder residual | DeepONet residual |

Figure 4: Visualization of AutoBasisEncoder DeepONet prediction results on *Darcy flow* dataset

### A.2.4 BASIS CONVERGENCE OF AUTOBASISENCODER

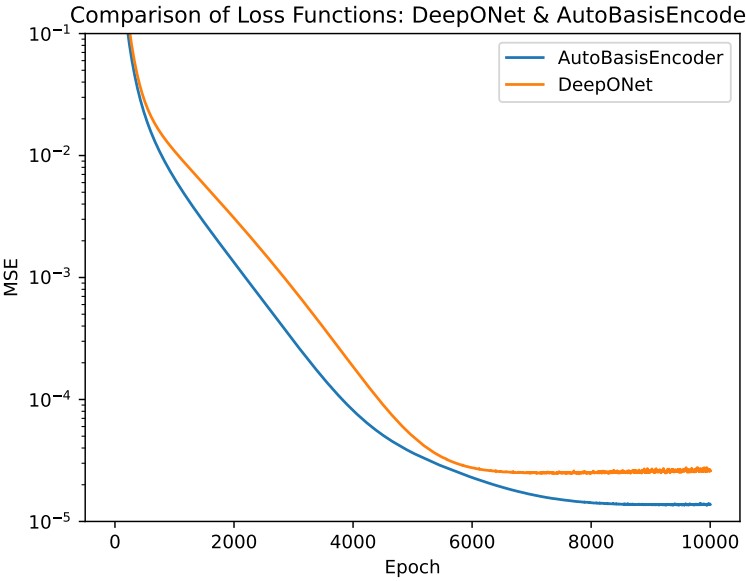

Figure 5: Convergence comparison of the reconstruction of functions of $\mathcal{V}$ using the basis learned by DeepONet and AutoBasisEncoder. The bases of functions are trained, and then used as pre-trained basis on the *Navier-Stokes* dataset at resolution 64.

