# OpenReview forum: "AutoBasisEncoder: Pre-trained Neural Field Basis via Autoencoding for Operator Learning"
_ICLR.cc/2024/Workshop/AI4DiffEqtnsInSci — AI4DiffEqtnsInSci @ ICLR 2024 Poster_

### Official Review · Reviewer_u5fE · 2024-02-25
**Review of AutoBasisEncoder: Pre-trained Neural Field Basis via Autoencoding for Operator Learning**

**Rating:** 5
**Confidence:** 3

**Review:**

The paper introduces a new framework for operator learning by separating the optimization of the basis coefficients from the basis functions in a pre-existing architecture, DeepONet. The authors first use an autoencoder to find the basis functions from the target space, which are then fixed to find the coefficients using the input functions, and show that this approach leads to the discovery of more expressive basis functions. The paper, for the most part, is clear - the idea is simple and is relevant to the field. A major weakness is that the autoencoding step to determine the basis functions requires knowledge of the target space at points (unlike DeepONet and POD-DeepONet which just uses locations to encode the basis functions). This limits its use for test examples where there is zero or limited access to the target space. There are also many typos in the paper.

---

### Official Review · Reviewer_ihpk · 2024-02-25

**Rating:** 5
**Confidence:** 3

**Review:**

DeepONet is a framework for operator learning that learns a basis of functions from data and represents functions as linear combinations of the bases. Both the basis functions and the network that predicts the coefficients of the linear combination are learned jointly. Following recent work (POD-DeepONet, BasisONet), this paper proposes AutoBasisEncoder that separates these into two steps: learning the basis function using an autoencoding objective and the learning the coefficients. Empirical results show improved performance on Darcy flow equation and the Navier-Stokes equation datasets over DeepONet and the ability to interpolate over different resolutions of data.

### Strengths
- The paper is easy to follow (although some discussion on related work can be improved; see weaknesses).
- The proposed approach improves over DeepONet on two datasets, highlighting the the effectiveness of disentangling basis learning from coefficient learning.

### Weaknesses
- The proposed method is simplistic in nature and the empirical analysis is also limited. This makes the overall contribution marginal, especially in the context of works such as BasisONet.
- Discussion on related work needs improvement. In particular, the differences from POD-DeepONet and BasisONet need to be highlighted more to better position the work.

### Questions
- Why did the authors not compare with BasisONet given its similarity to the proposed method?
- Why did the authors not use any interpolation techniques for POD-DeepONet in Table 2? Can they not be used?
- Why is POD-DeepONet missing from Fig. 3?

---

### Official Review · Reviewer_9T2Q · 2024-02-26
**This work presents an AutoBasisEncoder which is a framework designed for operator learning based on of set of functions optimized for the target function space and utilizes this pre-trained basis for efficient learning process.**

**Rating:** 7
**Confidence:** 3

**Review:**

This work presents an AutoBasisEncoder, which is a framework designed for operator learning based on of set of functions optimized for the target function space and utilizes this pre-trained basis for an efficient learning process.
The paper is well written and some conclusions underscore the potential of AutoBasisEncoder to enhance the landscape of operator learning frameworks

---

### Meta-Review · Area_Chair_pChU · 2024-03-01

**Recommendation:** Accept (Poster)

**Metareview:**

This paper introduces a novel framework designed for operator learning, mapping from one function to another. All reviewers agree unanimously. The concerns raised by reviewers can be address by enhancing the clarification of the paper. Therefore, I support the acceptance. However, author needs to address the concerns in the camera ready version

---

### Decision · Program_Chairs · 2024-03-01

Accept (Poster)